# Mutations that prevent phosphorylation of the BMP4 prodomain impair proteolytic maturation of homodimers leading to lethality in mice

Hyung-Seok Kim[1†], Mary L Sanchez[1†‡], Joshua Silva[1†§], Heidi L Schubert[2], Rebecca Dennis[1], Christopher P Hill[2], Jan L Christian[1,3]*

[1]Department of Neurobiology, University of Utah, Salt Lake City, United States; [2]Department of Biochemistry, University of Utah, Salt Lake City, United States; [3]Internal Medicine, Division of Hematology and Hematologic Malignancies, University of Utah, Salt Lake City, United States

*For correspondence:
jan.christian@neuro.utah.edu

[†]These authors contributed equally to this work

Present address: [‡]Center for Fundamental and Applied Microbiomics Arizona State University, Tempe, United States; [§]Cell and Molecular Biology, Duke University, Durham, United States

Competing interest: The authors declare that no competing interests exist.

## eLife Assessment

This **fundamental** work presents two clinically relevant BMP4 mutations that contribute to vertebrate development. The **compelling** evidence, both from wet lab and AI generated predictions, supports that the site-specific cleavage at the BMP4 pro-domain precisely regulates its function and provides mechanistic insight how homodimers and heterodimers behave differently. The work will be of broad interest to researchers working on growth factor signaling mechanisms and vertebrate development.

**Abstract** Bone morphogenetic protein4 (BMP4) plays numerous roles during embryogenesis and can signal either alone as a homodimer, or together with BMP7 as a more active heterodimer. BMPs are generated as inactive precursor proteins that dimerize and are cleaved to generate the bioactive ligand and inactive prodomain fragments. In humans, heterozygous mutations within the prodomain of BMP4 are associated with birth defects. We studied the effect of two of these mutations (p.S91C and p.E93G), which disrupt a conserved FAM20C phosphorylation motif, on ligand activity. We compared the activity of ligands generated from BMP4, BMP4$^{S91C}$, or BMP4$^{E93G}$ in *Xenopus* embryos and found that these mutations reduce the activity of BMP4 homodimers but not BMP4/7 heterodimers. We generated *Bmp4*$^{S91C}$ and *Bmp4*$^{E93G}$ knock-in mice and found that *Bmp4*$^{S91C/S91C}$ mice die by E11.5 and display reduced BMP activity in multiple tissues including the heart. Most *Bmp4*$^{E93G/E93G}$ mice die before weaning and *Bmp4*$^{-/E93G}$ mutants die prenatally with reduced or absent eyes, heart, and ventral body wall closure defects. Mouse embryonic fibroblasts (MEFs) isolated from *Bmp4*$^{S91C}$ and *Bmp4*$^{E93G}$ embryos show accumulation of BMP4 precursor protein, reduced levels of cleaved BMP ligand and reduced BMP activity relative to MEFs from wild type littermates. Because *Bmp7* is not expressed in MEFs, the accumulation of unprocessed BMP4 precursor protein in mice carrying these mutations most likely reflects an inability to cleave BMP4 homodimers, leading to reduced levels of ligand and BMP activity in vivo. Our results suggest that phosphorylation of the BMP4 prodomain is required for proteolytic activation of BMP4 homodimers, but not heterodimers.

## Introduction

BMPs are secreted molecules that were initially discovered as bone inducing factors and were subsequently shown to play numerous roles during embryogenesis (*Bragdon et al., 2011*). *Bmp2, 4, 5, 6,* and *7* are broadly expressed throughout embryogenesis, often in overlapping patterns (*Danesh et al.,*

2009). *Bmp2* and *Bmp4* play important and non-redundant developmental roles, as mice homozygous for null mutations in either gene null die during early development (*Winnier et al., 1995*; *Zhang and Bradley, 1996*). Although *Bmp5*, *Bmp6*, or *Bmp7* null homozygotes survive until birth or beyond, *Bmp5;Bmp7* and *Bmp6;Bmp7* double mutants die during embryogenesis, revealing functional redundancy within this subgroup (*Kim et al., 2001*; *Solloway and Robertson, 1999*).

BMPs are grouped into subfamilies based on sequence similarity and can signal as either homodimers or as heterodimers. The class I BMPs, BMP2, and BMP4, can heterodimerize with class II BMPs, consisting of BMPs 5–8 (*Guo and Wu, 2012*). Heterodimers composed of class I and II BMPs show a higher specific activity than homodimers. For example, BMP2/7, BMP4/7, and BMP2/6 heterodimers are significantly more potent than any homodimer in multiple assays (*Aono et al., 1995*; *Chauhan et al., 2024*; *Kaito et al., 2018*; *Valera et al., 2010*). In vivo, endogenous Bmp2/7 heterodimers are essential to establish the dorsoventral axis in fish (*Little and Mullins, 2009*) and *Drosophila* (*Shimmi et al., 2005*). More recent studies have shown that endogenous BMP4/7 and BMP2/7 heterodimers are required to generate full BMP activity in most or all tissues of mouse embryos (*Kim et al., 2019*).

The choice of whether a given BMP will form a homodimer or a heterodimer is made within the biosynthetic pathway. BMPs are made as inactive precursor proteins that dimerize and fold within the endoplasmic reticulum (ER) and are cleaved by members of the proprotein convertase family, such as furin, to generate the active, disulfide-bonded ligand and two prodomain monomers (*Bragdon et al., 2011*). Prodomains lack canonical signaling activity but are essential for ligand folding and dimerization, and can regulate subcellular trafficking, localization, and bioavailability of mature ligands (*Constam, 2014*; *Cui et al., 2001*; *Goldman et al., 2006*; *Gray and Mason, 1990*; *Harrison et al., 2011*; *Sengle et al., 2008*; *Tilak et al., 2014*). Structural studies show that some dimerized TGF-β superfamily precursors adopt a conformation whereby the prodomain of one monomer contacts the ligand domain of the opposite monomer (*Wang et al., 2016*; *Zhao et al., 2018*), suggesting that prodomains may play a key role in heterodimer formation and function.

Bmp4 and Bmp7 preferentially form heterodimers rather than either homodimer when co-expressed in *Xenopus*, and the Bmp4 prodomain is necessary and sufficient to generate properly folded, functional BMP4 homodimers and Bmp4/7 heterodimers (*Neugebauer et al., 2015*). Bmp4 is sequentially cleaved at two sites within the prodomain to generate an active ligand (*Cui et al., 2001*). An initial cleavage occurs adjacent to the ligand domain, and this generates a transient non-covalently associated prodomain/ligand complex (*Figure 1A*). A second cleavage at an upstream site generates two prodomain fragments (dark green and yellow boxes, *Figure 1A*) and dissociates the prodomain/ligand complex (*Degnin et al., 2004*). Sequential cleavage at both sites, and formation of the transient prodomain/ligand complex is essential to generate a stable, fully active ligand (*Goldman et al., 2006*; *Tilak et al., 2014*). By contrast, BMP7 is cleaved at a single site and the prodomains remain transiently and non-covalently associated with the ligand following cleavage. The type II BMP receptor then competes with the N-terminus of the prodomain for ligand binding and displaces the prodomain from the complex to allow for downstream signaling (*Sengle et al., 2008*). The same may be true for Bmp4/7 heterodimers, since both prodomains remain non-covalently attached following cleavage (*Figure 1B*; *Neugebauer et al., 2015*). The factors that drive formation of fully functional heterodimers or homodimers are unknown.

In humans, heterozygous mutations within the prodomain of *BMP4* are associated with a spectrum of ocular, brain, kidney, dental, and palate abnormalities (*Bakrania et al., 2008*; *Chen et al., 2007*; *Nixon et al., 2019*; *Reis et al., 2011*; *Schild et al., 2013*; *Suzuki et al., 2009*; *Weber et al., 2008*; *Yu et al., 2019*; *Zhang et al., 2009*). Two of the BMP4 prodomain missense mutations found in humans (c.271A>T, p.S91C and c.278A>G, p.E93G) disrupt a highly conserved phosphorylation motif (S-X-E/pS) (*Figure 1C*) that is recognized by the secretory pathway kinase Family with sequence similarity to 20C (FAM20C). FAM20C phosphorylates endogenous BMP4 at S91 within this motif (*Tagliabracci et al., 2015*) and BMP activity is decreased in ameloblasts and dental epithelium from *Fam20c* mutant mice (*Liu et al., 2018*; *Liu et al., 2020*). Mutations in human *FAM20C* cause the often-lethal disorder, Raine syndrome (*Simpson et al., 2007*). Some of the dental and bone defects observed in Raine patients can be accounted for by loss of phosphorylation of FAM20C substrates involved in biomineralization (*Faundes et al., 2014*) and of FGF23 (*Tagliabracci et al., 2014*). The etiology of other defects in these patients, such as cleft palate and craniofacial abnormalities, is unclear. Whether and how phosphorylation of the prodomain impacts BMP ligand formation or activity is unknown.

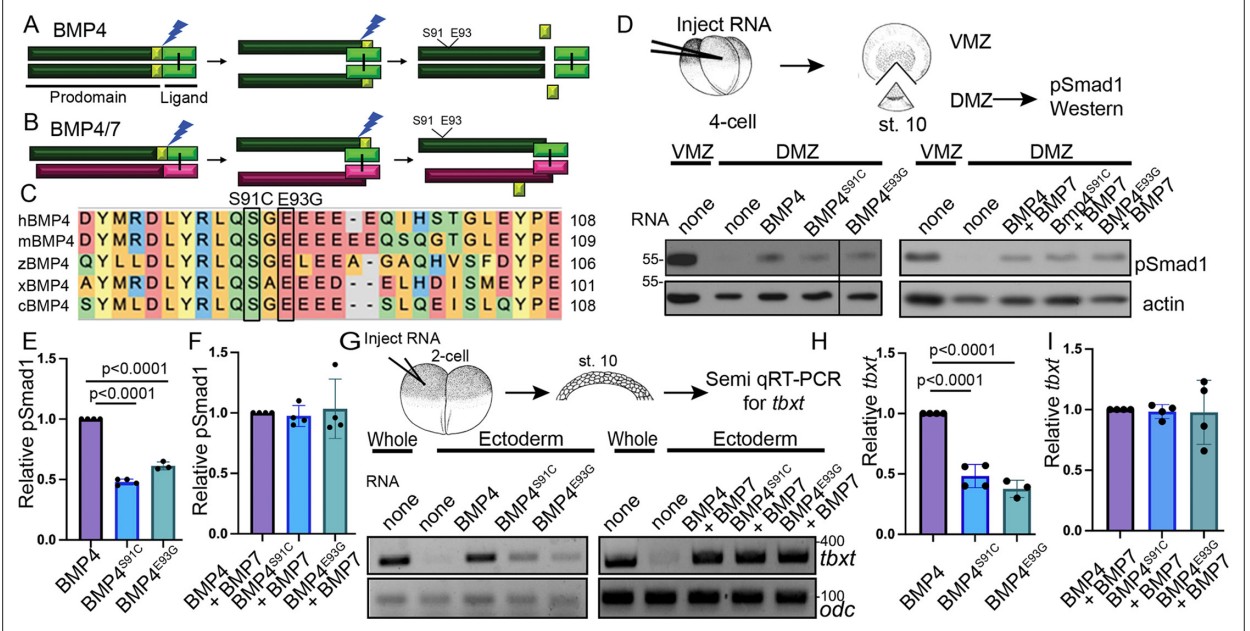

**Figure 1.** Point mutations predicted to interfere with phosphorylation of the BMP4 prodomain selectively interfere with BMP4 homodimer but not BMP4/7 heterodimer activity. Schematic illustrating sequential cleavage of BMP4 homodimers (**A**) and BMP4/7 heterodimers (**B**). BMP4 is sequentially cleaved at two sites to generate the mature ligand (light green) together with large (dark green) and small (yellow) prodomain fragments. (**C**) Sequence alignment of a portion of the prodomain of human (h), mouse (m), zebrafish (z), *Xenopus* (x), and chick (c) BMP4 to illustrate the conserved S-X-G FAM20C recognition motif. (**D**) RNA encoding wild type or point mutant forms of BMP4 were injected alone or together with BMP7 near the dorsal marginal zone (DMZ) of 4-cell embryos. DMZ and ventral marginal zone (VMZ) explants were isolated at stage 10 and pSmad1 levels were analyzed by immunoblot as illustrated. Blots were reprobed with actin as a loading control. Black bar indicates where a non-relevant intervening lane was removed using photoshop. (**E, F**) Quantitation of relative pSmad1 levels normalized to actin in at least three independent experiments (mean ± SD, data analyzed using an unpaired *t*-test). (**G**) RNA encoding wild type or point mutant forms of BMP4 were injected alone or together with BMP7 near the animal pole of 2-cell embryos. Ectodermal explants were isolated at stage 10 and *tbxt* levels were analyzed by semi-quantitative RT-PCR as illustrated. (**H, I**) Quantitation of relative *tbxt* levels normalized to *odc* in at least three independent experiments (mean ± SD, data analyzed using an unpaired *t*-test).

The online version of this article includes the following source data and figure supplement(s) for figure 1:

**Source data 1.** TIFF files containing original western blots for *Figure 1D*, indicating the relevant bands, explants, and deleted lane.

**Source data 2.** TIFF files containing original western blots for *Figure 1D*.

**Source data 3.** TIFF files containing original PCR scans for *Figure 1G*, indicating the relevant bands and injected RNAs.

**Source data 4.** TIFF files containing original PCR scans for *Figure 1G*, indicating injected RNAs.

**Figure supplement 1.** Ectopic expression of a phosphomimetic mutant form of BMP4 (BMP4S91D) generates more activity than BMP4S91C but less activity than native BMP4.

**Figure supplement 1—source data 1.** TIFF files containing original western blots for *Figure 1—figure supplement 1A*, indicating the relevant bands and genotypes.

**Figure supplement 1—source data 2.** TIFF files containing original western blots for *Figure 1—figure supplement 1A*.

**Figure supplement 2.** Point mutations predicted to interfere with phosphorylation of the BMP4 prodomain do not interfere with BMP4/7 heterodimer activity.

**Figure supplement 2—source data 1.** TIFF files containing original western blots for *Figure 1—figure supplement 2A*, indicating the relevant bands and genotypes.

**Figure supplement 2—source data 2.** TIFF files containing original western blots for *Figure 1—figure supplement 2A*.

**Figure supplement 2—source data 3.** TIFF files containing original PCR scans for *Figure 1—figure supplement 2*, indicating the relevant bands and injected RNAs.

**Figure supplement 2—source data 4.** TIFF files containing original PCR scans for *Figure 1—figure supplement 2*.

In the current studies, we analyzed the impact of p.S91C and p.E93G prodomain mutations on ligand function. We found that these mutations disrupt the formation of functional BMP4 homodimers, but not BMP4/7 heterodimers in ectopic expression assays. Mice carrying $Bmp4^{S91C}$ or $Bmp4^{E93G}$ knock in mutations show embryonic or enhanced postnatal lethality, respectively. MEFs isolated from mutant mice show reduced BMP activity and accumulation of uncleaved homodimeric BMP4 precursor proteins with a concomitant loss of cleaved ligand. Our results suggest that phosphorylation of the BMP4 prodomain is required for proteolytic activation of BMP4 homodimers, but not heterodimers.

## Results

### Point mutations predicted to interfere with phosphorylation of the BMP4 prodomain selectively interfere with BMP4 homodimer but not BMP4/7 heterodimer activity

Humans heterozygous for missense mutations p.S91C or p.E93G within the BMP4 prodomain (*Figure 1A, C*) display enhanced predisposition to colorectal cancer, micropthalmia, skeletal defects, brain abnormalities, cleft lip, and/or kidney dysgenesis (*Bakrania et al., 2008*; *Lubbe et al., 2011*; *Suzuki et al., 2009*; *Weber et al., 2008*), suggesting that these prodomain mutations interfere with the activity of BMP4 homodimers or BMP4/7 heterodimers. To test this possibility, we injected RNA (25 pg) encoding murine BMP4$^{HAMyc}$, BMP4$^{HAMycS91C}$, or BMP4$^{HAMycE93G}$ into *Xenopus* embryos near the dorsal marginal zone (DMZ) of 4-cell embryos. We have previously shown that these epitope tags do not interfere with the activity of BMP4 in vivo (*Tilak et al., 2014*). The DMZ was explanted from embryos at the early gastrula stage and immunoblots of DMZ extracts were probed for phos-phoSmad1,5,8 (hereafter shortened to pSmad1), which provides a direct read out for BMP activity (assay illustrated in *Figure 1D*). Levels of pSmad1 are high in the VMZ, where endogenous Bmp ligands are expressed, but low in the DMZ, where BMP inhibitors are expressed. Ectopic BMP4$^{S91C}$ or BMP4$^{E93G}$ induced significantly less pSmad1 than did wild type BMP4 (*Figure 1D, E*), demonstrating that these point mutations interfere with BMP4 homodimer activity. We also tested whether a putative phosphomimetic mutant (BMP4$^{S91D}$) would rescue ligand activity relative to BMP4$^{S91C}$. This mutant serves as a control for the possibility that the p.S91C mutant causes phosphorylation independent defects in protein function due to aberrant disulfide bonding with the ectopic cysteine residue, as suggested by others (*Tabatabaeifar et al., 2009*). Consistent with this possibility, pSmad1 activity was significantly higher in DMZ explants expressing BMP4$^{S91D}$ than in those expressing BMP4$^{S91C}$ (*Figure 1—figure supplement 2A, B*). Notably, pSmad1 activity induced by BMP4$^{S91D}$ remained lower than that induced by wild type BMP4 (*Figure 1—figure supplement 1A, B*), conceivably due to other regulatory or conformational constraints on Ser91 or to the inability of this amino acid substitution to mimic the endogenous phosphorylated state. We used a second ectopic expression assay to verify results. RNA encoding wild type or mutant BMP4 (25 pg) was injected near the animal pole of 2-cell *Xenopus* embryos, ectoderm was explanted at the early gastrula stage and expression of the BMP target gene, *tbxt*, was analyzed by semi-quantitative (semi-q) RT-PCR. *tbxt* levels were significantly lower in explants from embryos expressing BMP4$^{S91C}$ or BMP4$^{E93G}$ relative to those expressing wild type BMP4 (*Figure 1G, H*). These findings are consistent with the possibility that phosphorylation of S91 is required to generate fully active BMP4 homodimers.

We then repeated these assays in embryos co-injected with RNA encoding wild type or mutant BMP4 (12.5 pg) together with BMP7 (12.5 pg). Previous studies have shown that BMP7 homodimers have weak or no activity relative to BMP4 homodimers, that BMP4 and BMP7 preferentially generate heterodimers over either homodimer when co-expressed in *Xenopus*, and that BMP4/7 heterodimers generate significantly more BMP activity than either homodimer in *Xenopus* ectodermal and mesodermal assays (*Neugebauer et al., 2015*; *Nishimatsu and Thomsen, 1998*; *Suzuki et al., 1997*; *Figure 1—figure supplement 2*). There was no significant difference in the levels of pSmad1 (*Figures 1 and 2*) or *tbxt* (*Figures 1 and 2*) induced in embryos co-injected with BMP7 together with either point mutant form of BMP4 relative to embryos co-injected with BMP7 and wild type BMP4. This suggests that the S91C and E93G prodomain point mutations selectively interfere with BMP4 homodimer but not BMP4/7 heterodimer activity.

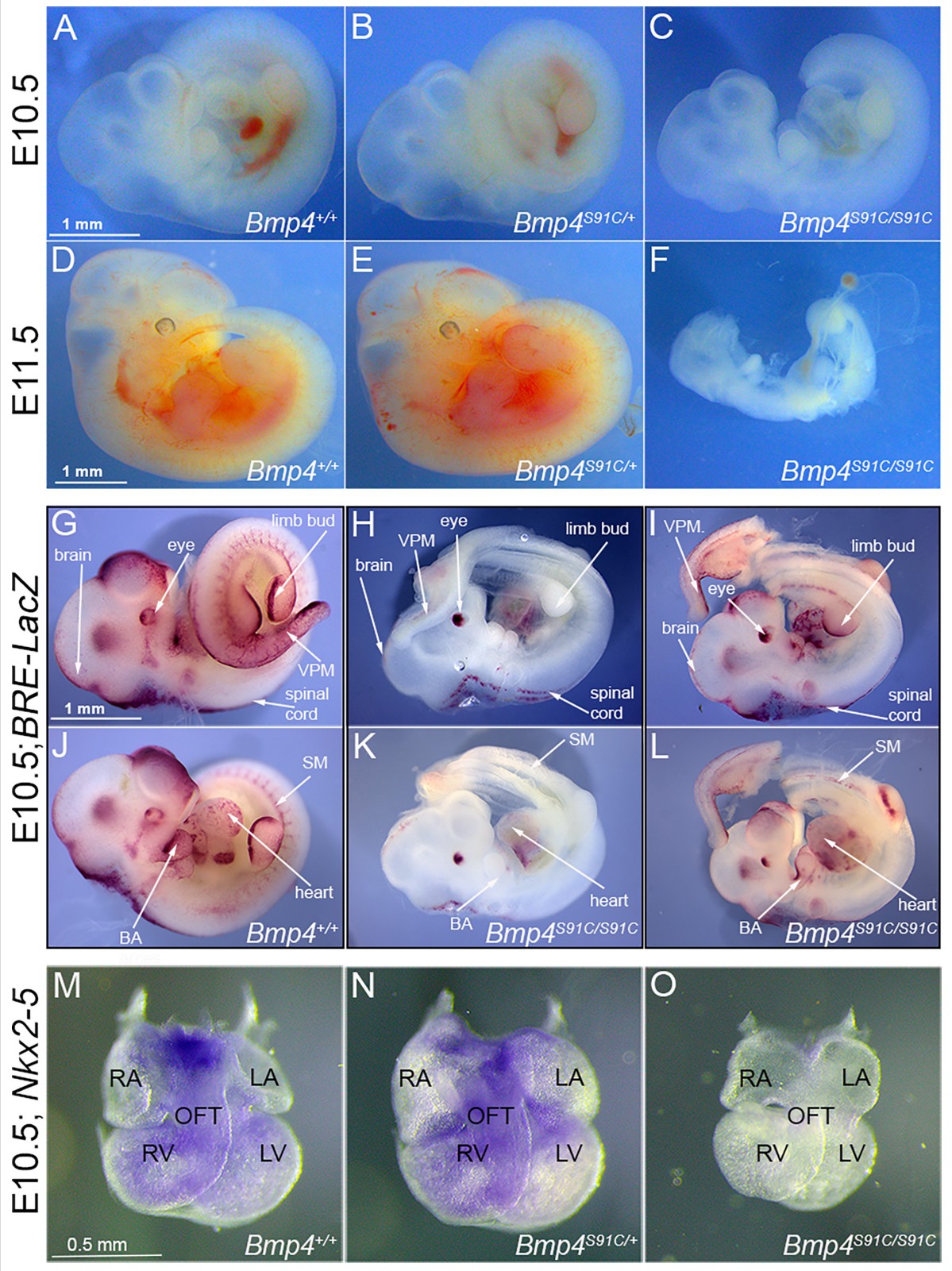

**Figure 2.** Bmp4$^{S91C}$ homozygotes die during mid-embryogenesis and show reduced BMP activity in multiple tissues (**A–F**). Photograph of E10.5 or E11.5 wild type (**A, D**) or mutant (**B, C, E, F**) littermates. Scale bars in panels A and D apply across each row. (**G–J**) E10.5 wild type (**G, J**) or Bmp4$^{S91C/S91C}$ mutant littermates (**H, I, K, L**) carrying a BRE-LacZ transgene were stained for β-galactosidase activity to detect endogenous BMP pathway activation. (**G–I**) show right side and (**J–L**) the left side of the same embryos. Embryos from a single litter were stained for an identical time under identical

*Figure 2 continued on next page*

*Figure 2 continued*

conditions. A minimum of three embryos of each genotype were examined and results shown were reproduced in all. VPM: ventral posterior mesoderm, BA: branchial arches, SM: somitic mesoderm. Scale bar corresponds to 1 mm in all panels. Expression of *Nkx2.5* was analyzed by whole mount in situ hybridization in E10.5 wild type (**M**) and mutant (**N, O**) littermates. Photographs of hearts dissected free of embryos are shown. Scale bar in M applies across the row. RA: right atrium, LA: left atrium, RV: right ventricle, LV: left ventricle, OFT: outflow tract.

The online version of this article includes the following figure supplement(s) for figure 2:

**Figure supplement 1.** Bmp4S91C homozygotes show reduced BMP activity in the heart and ventral posterior mesoderm at E9.5.

## Bmp4$^{S91C}$ homozygotes die during mid-embryogenesis and show reduced BMP activity in multiple tissues

To ask whether S91C and E93G prodomain missense mutations interfere with endogenous BMP4 activity in vivo, we introduced nucleotide mutations to encode single amino acid changes (p.S91C or p.E93G) along with sequence encoding an HA epitope tag into the prodomain of the *Bmp4* allele in mice (*Bmp4$^{S91C}$* and *Bmp4$^{E93G}$*). We have previously generated control mice that carry an HA epitope tag at the same position in the wild type *Bmp4* locus, as well as a myc tag in the ligand domain (*Bmp4$^{HAMyc}$*). These mice are adult viable and show no visible defects (*Tilak et al., 2014*).

*Bmp4$^{S91/+}$* mice were intercrossed to determine whether homozygotes were adult viable. *Bmp4$^{S91C}$* homozygotes were not recovered at weaning (*Table 1A*). We then established timed matings to determine when during embryogenesis death occurred. *Bmp4$^{S91C/S91C}$* mutants were recovered at the predicted Mendelian frequency at embryonic day (E)9.5 and E10.5 (*Table 2A*, *Supplementary file 1C,D*) but were slightly smaller than littermates at E10.5 (*Figure 2A–C*) and were absent or resorbing by E11.5 (*Figure 2D–F*, *Table 2A*, *Supplementary file 1A,B*).

**Table 1.** Progeny from *Bmp4$^{S91C/+}$* or *Bmp4$^{E93G/+}$* intercrosses, or from *Bmp4$^{-/+}$* and *Bmp4$^{E93G/+}$* crosses at P28.

**A. Progeny from *Bmp4$^{S91C/+}$* intercrosses**

| Sex | *Bmp4$^{+/+}$* | *Bmp4$^{S91C/+}$* | *Bmp4$^{S91C/S91C}$* | n | p |
|---|---|---|---|---|---|
| Both | 12 (14) | 44 (28) | 0 (14) | 56 | 1E−05 |
| Male | 7 (8) | 23 (15) | 0 (8) | 30 | 0.0027 |
| Female | 6 (7) | 20 (13) | 0 (7) | 26 | 0.0034 |

**B. Progeny from *Bmp4$^{E93G/+}$* intercrosses**

| Sex | *Bmp4$^{+/+}$* | *Bmp4$^{E9G3G/+}$* | *Bmp4$^{E93G/E93G}$* | n | p |
|---|---|---|---|---|---|
| Both | 35 (31) | 74 (63) | 16 (31) | 125 | 0.007 |
| Male | 19 (16) | 31 (31) | 12 (16) | 62 | 0.454 |
| Female | 17 (16) | 44 (32) | 4 (16) | 63 | 0.002 |

**C. Progeny from *Bmp4$^{-/+}$* × *Bmp4$^{E93G/+}$* crosses**

| Sex | *Bmp4$^{+/+}$* | *Bmp4$^{E9G3G/+}$* | *Bmp4$^{-/+}$* | *Bmp4$^{E93G/-}$* | n | p |
|---|---|---|---|---|---|---|
| Both | 22 (15) | 22 (15) | 17 (15) | 0 (15) | 61 | 1E−04 |
| Male | 10 (8) | 9 (8) | 11 (8) | 0 (8) | 30 | 0.016 |
| Female | 13 (8) | 11 (8) | 7 (8) | 0 (8) | 31 | 0.005 |

**D. Progeny from *Bmp4$^{S91C/+}$* × *Bmp4$^{E93G/+}$* crosses**

| Sex | *Bmp4$^{+/+}$* | *\*Bmp4$^{S91C/+}$* or *Bmp4$^{E9G3G/+}$* | *Bmp4$^{ES91C/E93G}$* | n | p |
|---|---|---|---|---|---|
| Both | 23 (17) | 39 (34) | 7 (17) | 69 | 0.014 |
| Male | 9 (9) | 23 (18) | 4 (9) | 36 | 0.125 |
| Female | 14 (8) | 6 (17) | 3 (8) | 33 | 0.025 |

(A–D) Numbers of observed and expected (in parenthesis) mice of each genotype listed in the top row are indicated. Genotypes reported at P28. The p value is based on $X^2$ test.

(D) *Genotyping protocol does not distinguish between *Bmp4$^{S91C/+}$* or *Bmp4$^{E9G3G/+}$* alleles and thus heterozygotes are pooled.

**Table 2.** Progeny from $Bmp4^{S91C/+}$ intercrosses, or from $Bmp4^{-/+}$ and $Bmp4^{E93G/+}$ intercrosses at embryonic ages.

**A. Progeny from $Bmp4^{S91C/+}$ intercrosses**

| Age | $Bmp4^{+/+}$ | $Bmp4^{S91C/+}$ | $Bmp4^{S91C/S91C}$ | n | p |
|---|---|---|---|---|---|
| E13.5 | 22 (19) | 55 (39) | 0 (19) | 77 | 1E−05 |
| E11.5 | 21 (14) | 35 (28) | 0 (14) | 56 | 2E−04 |
| E10.5 | 43 (51) | 18 (102) | 43 (51) | 204 | 0.211 |
| E9.5 | 7 (12) | 31 (24) | 9 (12) | 47 | 0.084 |

**B. Progeny from $Bmp4^{E93G/+}$ × $Bmp4^{-/+}$ crosses**

| Age | $Bmp4^{+/+}$ | $Bmp4^{E93G/+}$ | $Bmp4^{-/+}$ | $Bmp4^{-/E93G}$ | n | p |
|---|---|---|---|---|---|---|
| E13.5–14.5 | 17 (17) | 13 (17) | 19 (17) | 17 (17) | 66 | 0.77 |
| E11.5–12.5 | 11 (10) | 4 (10) | 12 (10) | 14 (10) | 41 | 0.14 |
| E10.5 | 7 (7) | 10 (7) | 7 (7) | 5 (7) | 29 | 0.62 |

(A, B) Numbers of observed and expected (in parenthesis) embryos of each genotype listed in the top row are indicated. The p value is based on $X^2$ test.

To determine whether and where BMP activity is reduced in $Bmp4^{S91C}$ homozygotes, we analyzed BMP activity in *BRE:LacZ* transgenic embryos at E10.5. This transgene contains a BMP-responsive element coupled to LacZ, which serves as an in vivo reporter of BMP signaling downstream of all endogenous BMP ligands (*Monteiro et al., 2004*). X-GAL staining of $Bmp4^{+/+}$;*BRE:LacZ* embryos revealed strong endogenous BMP activity in the brain and spinal cord, eye, branchial arches, limb buds, heart, somitic mesoderm, and ventroposterior mesoderm (VPM) (*Figure 2G, J*). $Bmp4^{S91C/S91C}$;*BRE:LacZ* mutants were smaller and exhibited a severe (*Figure 2H, K*, one out of six embryos) or modest (*Figure 2I, L*; five out of six embryos) reduction in BMP activity in all tissues except the eye. Notably, LacZ staining was also reduced in the heart and VPM of $Bmp4^{S91C/S91C}$;*BRE:LacZ* embryos relative to wild type littermates at E9.5 (*Figure 2—figure supplement 1*).

We also examined expression of the BMP target gene *Nkx2-5* in the heart of wild type and $Bmp4^{S91C}$ mutant littermates using whole mount in situ hybridization. At E10.5 expression of Nkx2-5 was reduced (1/3) or nearly absent (2/3) in hearts of $Bmp4^{S91C/S91C}$ embryos relative to littermates (*Figure 2M–O*). Although mutant hearts were smaller than littermates, they showed grossly normal patterning of atria, ventricles, and outflow tract (*Figure 2M–O*).

$Bmp4^{E93G/E93G}$ female mice are underrepresented at weaning and $Bmp4^{-/E93G}$ mutants die during late gestation with defects in ventral body wall closure, small eyes, and heart defects.

$Bmp4^{E93/+}$ mice were intercrossed to determine viability. $Bmp4^{E93G}$ homozygotes were underrepresented at weaning, and this could be entirely accounted for by an underrepresentation of female but not male $Bmp4^{E93G/E93G}$ mice (*Table 1B*). $Bmp4^{E93G/E93G}$ male mice appeared grossly normal at weaning,

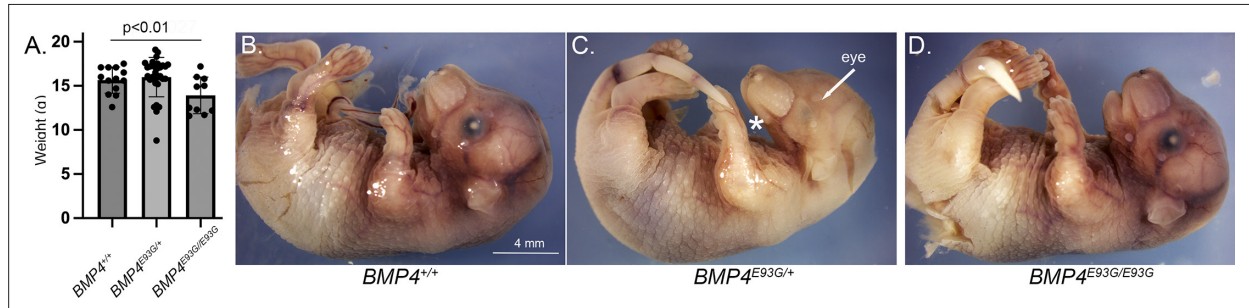

**Figure 3.** $Bmp4^{E93G}$ homozygotes are smaller than wild type littermates and a subset of mutants have eye and/or craniofacial defects. (**A**) Weights of male wild type and $Bmp4^{E93G}$ heterozygous and homozygous mutant littermates at P28 (mean ± SD, data analyzed using an unpaired *t*-test). (**B–D**) Photographs of E17.5 wild type and mutant littermates. A subset of $Bmp4^{E93G}$ heterozygotes show craniofacial defects such as a small mandible (asterisk) and small or absent eyes (arrow) at low frequency (n = 1/12).

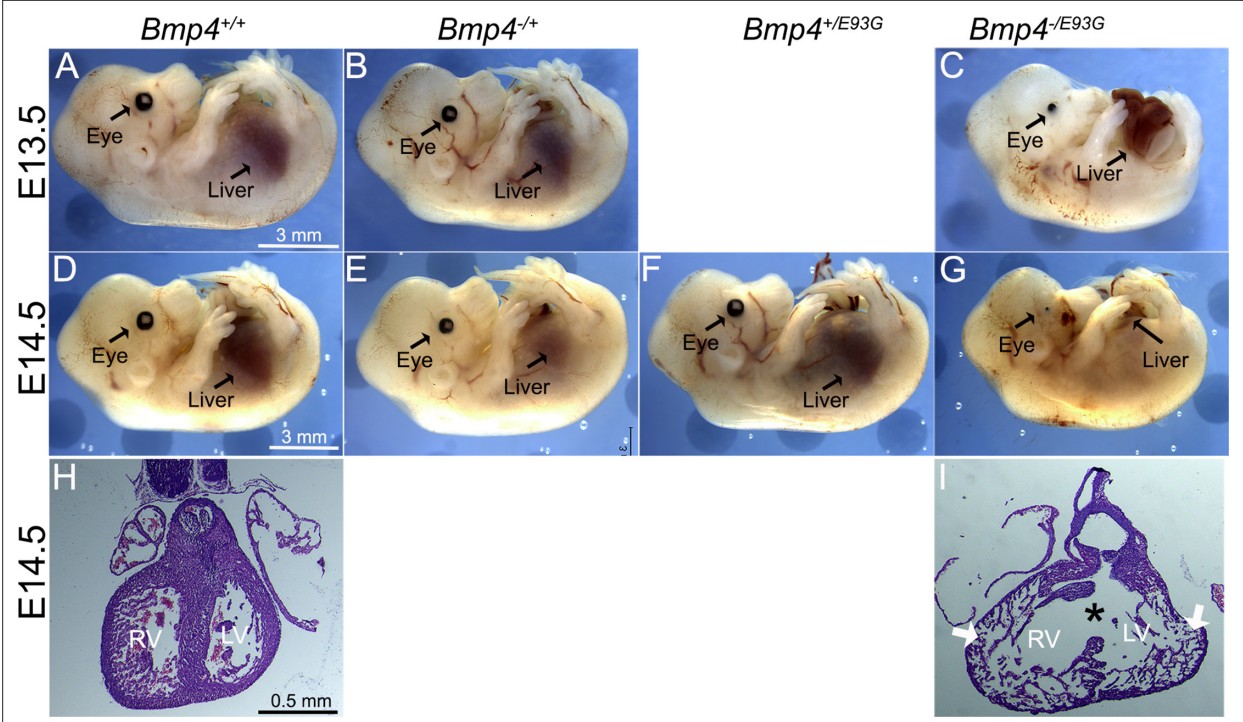

**Figure 4.** *Bmp4$^{-/E93G}$* mutants die during embryogenesis with defects in ventral body wall closure, small or absent eyes and heart defects. Photographs of E13.5 (**A–C**) or E14.5 (**D–G**) wild type (**A, D**) or mutant (**B, C, E–G**) littermates. The position of the liver inside (**A–F**) or outside (**C, G**) of the abdomen is indicated. Hematoxylin and eosin-stained coronal sections of hearts dissected from E14.5 wild type (**H**) or *Bmp4$^{-/E93G}$* littermates (**I**). Asterisk denotes ventricular septal defect and arrowheads indicate thin, non-compacted ventricular wall (**I**). Scale bar applies across each row.

The online version of this article includes the following source data for figure 4:

**Source data 1.** TIFF files containing original western blots for **Figure 1D**, indicating the relevant bands, explants and deleted lane.

although they weighed slightly less than wild type littermates (**Figure 3A**). Unilateral or bilateral microphthalmia, anopthalmia and/or craniofacial defects were observed with low frequency in *Bmp4$^{E93G}$* heterozygotes at late gestation (**Figure 3B–D**) (*n* = 1/12) or at weaning (*n* = 6/74). These defects have been previously reported in *Bmp4* hypomorphic mice (**Bonilla-Claudio et al., 2012**; **Furuta and Hogan, 1998**; **Goldman et al., 2006**) and in humans heterozygous for the *BMP4$^{E93}$* prodomain mutation (**Bakrania et al., 2008**).

To ask whether a single allele of *Bmp4$^{E93G}$* is sufficient to support viability, we intercrossed mice heterozygous for a null allele of *Bmp4* (*Bmp4$^{-/+}$*) with *Bmp4$^{E93G}$* heterozygotes. *Bmp4$^{-/E93G}$* compound mutants were not recovered at weaning (**Table 1C**) but were recovered at the predicted Mendelian frequency through E14.5 (**Table 2B**, **Supplementary file 2**). At E13.5–14.5, all *Bmp4$^{-/E93G}$* mutants were slightly runted relative to littermates and had defects in ventral body wall closure in which the liver was partially or fully externalized (*n* = 16/16) (**Figure 4A–G**). Most *Bmp4$^{-/E93G}$* embryos also had small or absent eyes (*n* = 15/16) (**Figure 4C, G**). A subset of mice heterozygous for the null allele of *Bmp4* also had small or absent eyes (2/19) as previously reported (**Dunn et al., 1997**). Histological examination of hearts dissected from *Bmp4$^{-/E93G}$* mutants at E14.5 revealed highly trabeculated ventricular walls that failed to undergo compaction (3/3) and ventricular septal defects (2/3) (VSDs) (**Figure 4I**) that were never observed in wild type embryos (**Figure 4H**). A similar spectrum of abnormalities including small or absent eyes, failure to close the ventral body wall and VSDs have previously been reported in mice with reduced dosage of *Bmp4* (**Furuta and Hogan, 1998**; **Goldman et al., 2009**; **Kim et al., 2019**; **Uchimura et al., 2009**).

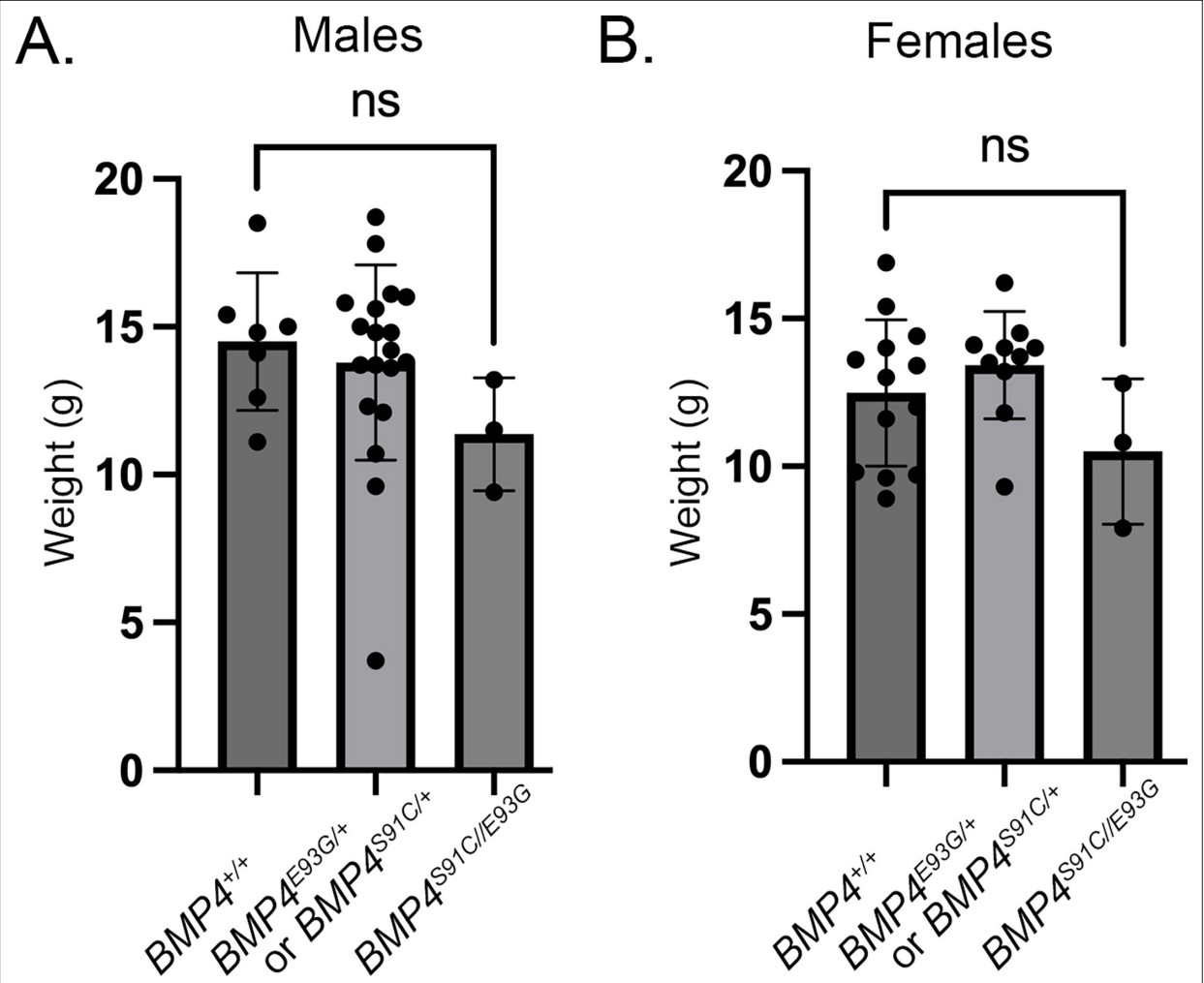

**Figure 5.** *Bmp4^{S91C/E93G}* compound heterozygotes are slightly smaller than heterozygous or wild type littermates. Weights of male (**A**) and female (**B**) wild type, heterozygous and compound heterozygous mutant littermates at P28 (mean ± SD, data analyzed using an unpaired *t*-test). Genotyping protocol does not distinguish between the *Bmp4^{S91C/+}* or *Bmp4^{E9G3/+}* allele and thus weights of *Bmp4^{S91C/+}* or *Bmp4^{E9G3/+}* heterozygotes are reported together.

## *Bmp4^{S91C/E93G}* female compound heterozygotes are underrepresented at weaning

*Bmp4^{S91/+}* and *Bmp4^{E93G/+}* mice were intercrossed to determine whether compound heterozygotes were adult viable. *Bmp4^{S91C/E93G}* mice were underrepresented at weaning, and this could be entirely accounted for by an underrepresentation of female but not male compound mutants (***Table 1D***). *Bmp4^{S91C/E93G}* mice appeared grossly normal at weaning, although they weighed slightly, but not significantly, less than littermates (***Figure 5***). The late lethality of female, but not male *Bmp4^{S91C/E93G}* compound heterozygotes and *Bmp4^{E93G}* homozygotes, along with their grossly normal phenotype contrasts with the early (E11) lethality and more severe phenotypic defects of *Bmp4^{S91C}* homozygotes. These results raise the possibility that loss of FAM20C-mediated phosphorylation of the BMP4 prodomain fully accounts for the late lethality of *Bmp4^{E93G/E93G}* and *Bmp4^{S91C/E93G}* mutants whereas the p.S91C mutation may cause additional, FAM20C-independent defects in BMP4 function when encoded on both alleles of *Bmp4*.

## E93G and S91C mutations lead to accumulation of BMP4 precursor protein and reduced levels of cleaved ligand and pSmad1 in vivo

To ask if BMP activity and/or protein levels are reduced in *Bmp4^{S91C}* or *Bmp4^{E93G}* mutants, we used immunoblot analysis to compare levels of pSMAD1 and BMP4 in E10.5 embryonic lysates isolated

from wild type or mutant littermates. Levels of BMP4 precursor protein were significantly higher, and levels of pSMAD1 and of cleaved BMP4 ligand were significantly lower in E10.5 lysates from $Bmp4^{S91C}$ homozygotes compared to wild type controls (**Figure 6A–D**). No significant differences in levels of pSMAD1, BMP4 precursor or BMP4 ligand were observed in E10.5 lysates from $Bmp4^{E93G}$ mutants compared to wild type controls (**Figure 6E–H**), or in wild type embryos carrying HA and Myc epitope tags compared to untagged controls (**Figure 6—figure supplement 1**). When we repeated this analysis in E13.5 mouse embryonic fibroblasts (MEFs) isolated from wild type or mutant littermates, we found that pSMAD1 levels were significantly reduced in $Bmp4^{S91C}$ heterozygotes (**Figure 6I, J**) and in $Bmp4^{E93G}$ homozygotes (**Figure 6M, N**) compared to wild type controls. In addition, levels of BMP4 precursor protein were increased and levels of cleaved ligand were reduced in $Bmp4^{S91C}$ heterozygotes (**Figure 6I, K, L**) and in $Bmp4^{E93G}$ heterozygotes or homozygotes (**Figure 6M, O, P**) compared to wild type controls. Thus, in E13.5 MEFs, p.E93G and p.S91C mutations in the prodomain of BMP4 interfere with proteolytic maturation of BMP4, leading to accumulation of unprocessed precursor protein and reduced levels of cleaved ligand and BMP activity in vivo.

Notably, in lysates from embryos heterozygous or homozygous for the S91C or E93G mutation, a second more slowly migrating BMP4 precursor band was observed (**Figure 6A, E**, arrow) in a subset of experiments. This band was never present in lysates from wild type littermates or in lysates from wild type embryos carrying HA and Myc epitope tags (**Figure 6—figure supplement 1**), indicating that the shift in migration is not due to the presence of epitope tags. The BMP4 precursor band in extracts of MEFs isolated from $Bmp4^{S91C/+}$ embryos also migrated more slowly than that in MEFs isolated from wild type littermates (**Figure 6I**). Treatment of proteins isolated from embryos or MEFs with calf intestinal phosphatase or treatment with deglycosylases to remove $O$- or $N$-linked glycosylation did not normalize the migration of this band. Thus, the retarded migration is unlikely to reflect differences in glycosylation or phosphorylation of mutant precursor protein relative to wild type.

## BMP4, BMP4$^{E93G}$, and BMP4$^{S91C}$ precursor proteins are $O$- and $N$-glycosylated and exit the ER

FAM20C-mediated phosphorylation of FGF23, like that of BMP4, is required for furin-mediated cleavage (**Wang et al., 2012**). Mechanistically, phosphorylation of FGF23 prevents $O$-glycosylation of a nearby residue, and $O$-glycosylation sterically blocks the furin cleavage site (**Tagliabracci et al., 2014**). To test if a similar mechanism accounts for lack of cleavage of BMP4$^{E93G}$ or BMP4$^{S91C}$, MEFs from E13.5 wild type, $Bmp4^{E93G/E93G}$ or $Bmp4^{S91C/+}$ mice were incubated in the presence or absence of $O$-glycosidase and neuraminidase and BMP4 precursor protein was analyzed on immunoblots. Treatment with these deglycosylases led to a more rapid migration of the BMP4 precursor protein, indicating that BMP4 is $O$-glycosylated, but there was no difference in the migration of wild type or point mutation precursors in the presence or absence of deglycosylases (**Figure 7A**), suggesting that phosphorylation does not alter $O$-glycosylation.

Another mechanism by which FAM20C-mediated phosphorylation might regulate cleavage of BMP4 is if it is required for the precursor protein to adopt its native folded conformation so that it is able to exit the ER. We analyzed $N$-linked glycosylation of endogenous wild type and mutant precursor proteins present in cultured MEFs. Immature $N$-linked carbohydrate residues that are transferred onto proteins in the ER are sensitive to digestion with Endoglycosidase H (Endo H), but when further modified in the Golgi they become Endo H resistant but remain sensitive to digestion with peptide $N$-glycosidase F (PNGase). Thus, Endo H resistance/PNGase sensitivity is a hallmark of proteins that are properly folded and able to traffic from the ER into the Golgi. As shown in **Figure 7B**, Endo H-sensitive (asterisks) and Endo H-resistant/PNGase F-sensitive (arrowheads) forms of wild type and mutant precursors were detected in MEFs. We have previously shown that high mannose, Endo H-sensitive carbohydrates are retained at one or more glycosylation site(s) on BMP4 ligand homodimers even after BMP4 has folded, trafficked through the trans-Golgi network (TGN) and been cleaved (**Degnin et al., 2004**). This explains why partial EndoH sensitivity is observed in both wild type and mutant precursors that have exited the ER. Collectively, our findings are consistent with a model in which FAM20C-mediated phosphorylation of the BMP4 prodomain is not required for folding or exit of the precursor protein from the ER, but is required for proprotein convertase recognition and/or for trafficking to post-TGN compartment(s) where BMP4 is cleaved (**Tilak et al., 2014**).

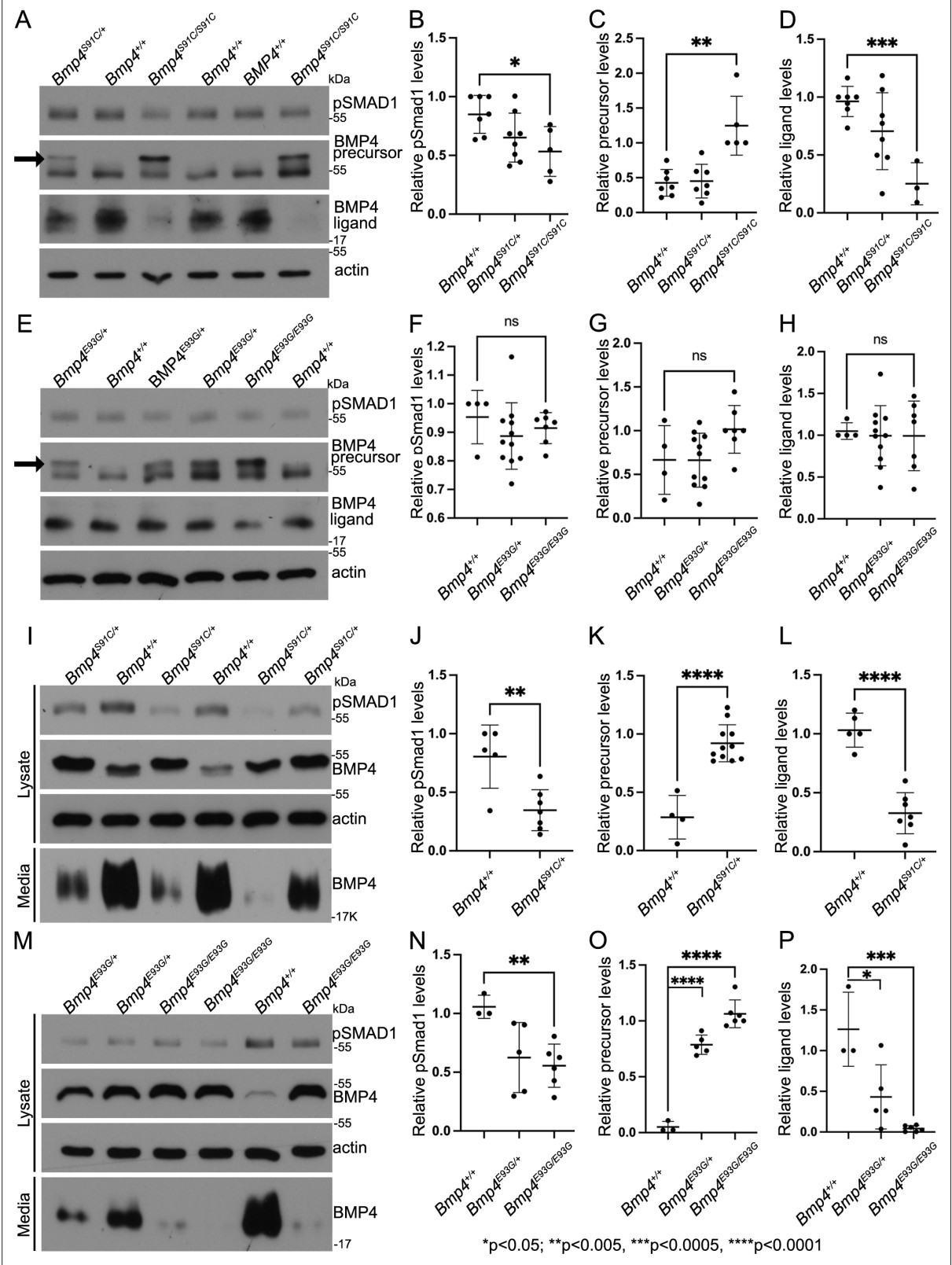

**Figure 6.** E93G and S91C mutations lead to accumulation of BMP4 precursor protein and reduced levels of cleaved ligand and pSmad1 in vivo. Levels of pSmad1 and BMP4 were analyzed in protein lysates isolated from E10.5 *Bmp4^{S91C}* (**A–D**) or *Bmp4^{E93G}* (**E–H**) homozygotes, heterozygotes, or wild type littermates. Arrows in panels A and E denote a more slowly migrating BMP4 precursor protein band detected in lysates from mutant, but not wild type embryos. (**I–P**) Levels of pSmad1, BMP4 precursor protein, and cleaved BMP4 ligand were analyzed in cell lysates and conditioned media of mouse

*Figure 6 continued on next page*

*Figure 6 continued*

embryonic fibroblasts (MEFs) isolated from E13.5 *Bmp4^S91C^* heterozygotes or wild type littermates (**I–L**) or from *Bmp4^E93G^* homozygotes, heterozygotes, or wild type littermates (**M–P**). Representative blots (**A, E, I, K**) and quantitation of protein levels normalized to actin (mean ± SD, data analyzed using an unpaired *t*-test) (**B–D, F–H, J–L, N, O**) are shown.

The online version of this article includes the following source data and figure supplement(s) for figure 6:

**Source data 1.** TIFF files containing original western blots for *Figure 6A*, indicating the relevant bands and genotypes.

**Source data 2.** TIFF files containing original western blots for *Figure 6A*.

**Source data 3.** TIFF files containing original western blots for *Figure 6E*, indicating the relevant bands and genotypes.

**Source data 4.** TIFF files containing original western blots for *Figure 6E*.

**Source data 5.** TIFF files containing original western blots for *Figure 6I*, indicating the relevant bands and genotypes.

**Source data 6.** TIFF files containing original western blots for *Figure 6I*.

**Source data 7.** TIFF files containing original western blots for *Figure 6M*, indicating the relevant bands and genotypes.

**Source data 8.** TIFF files containing original western blots for *Figure 6M*.

**Figure supplement 1.** Levels of BMP4 precursor protein, cleaved ligand and pSMAD1 are unchanged in MEFs isolated from Bmp4+/+ embryos carrying epitope tags relative to untagged littermates.

**Figure supplement 1—source data 1.** TIFF files containing original western blots for *Figure 6—figure supplement 1A*, indicating the relevant bands and genotypes.

**Figure supplement 1—source data 2.** TIFF files containing original western blots for *Figure 6—figure supplement 1A*.

## Discussion

Humans heterozygous for p.S91C or p.E93G point mutations within the prodomain of BMP4 show congenital birth defects and/or enhanced predisposition to colorectal cancer (*Bakrania et al., 2008*; *Lubbe et al., 2011*; *Suzuki et al., 2009*; *Weber et al., 2008*). These mutations disrupt a conserved -S-X-E- motif required for FAM20C-mediated phosphorylation of Ser91 suggesting that phosphorylation of the prodomain, which itself lacks biologic activity, is required for ligand activity. Here, we show that these mutations lead to reduced activity of BMP4 homodimers but not BMP4/7 heterodimers in *Xenopus* assays. Furthermore, in mice carrying either of these mutations proteolytic maturation of endogenous BMP4 is impaired in MEFs isolated at E13.5, leading to reduced BMP activity. Our results suggest that phosphorylation of the BMP4 prodomain by FAM20C is essential for furin to cleave the precursor protein to generate active BMP4 homodimers.

Our findings demonstrate that *Bmp4^S91C^* and *Bmp4^E93G^* are hypomorphic alleles, both of which carry mutations predicted to prevent FAM20C-mediated phosphorylation, but that *Bmp4^S91C^* mutants have a more severe loss of function. It is likely that the p.S91C mutation leads to additional defects in protein function independent of those caused by loss of phosphorylation. These are likely due to deleterious effects of introducing an ectopic cysteine residue into the precursor protein. BMP4 folds into a cystine knot containing three stereotypical intramolecular disulfide bonds that stabilize ligand monomers and a single intermolecular disulfide bond that stabilizes the dimer (*Schwarz, 2017*). The addition of an ectopic cysteine residue may lead to aberrant intramolecular or intermolecular disulfide bonds in the precursor protein, as suggested by biochemical analysis of BMP4^S91C^ homodimers expressed in cultured mammalian cells (*Tabatabaeifar et al., 2009*). This interpretation is consistent with our finding that *Bmp4^S91C/E93G^* compound heterozygotes have a less severe phenotype than *Bmp4^S91C/S91C^* mice, and with our observation that ectopically expressed BMP4^S91D^ generates significantly higher activity in *Xenopus* embryos than does BMP4^S91C^.

Phenotypic and biochemical analysis of *Bmp4^E93G^* and *Bmp4^S91C^* mice is consistent with a selective loss of BMP4 homodimers rather than BMP4/7 heterodimers. We have previously shown that mice homozygous for a mutation (*Bmp7^R-GFlag^*) that disrupts the formation of functional BMP2/7 and BMP4/7 heterodimers, or mice compound heterozygous for the *Bmp7^R-GFlag^* and *Bmp4* null alleles die by E11.5 (*Kim et al., 2019*). By contrast, *Bmp4^E93G/E93G^* and *Bmp4^S91C/E93G^* are grossly normal at E13.5, although a subset of mice dies during late gestation or early postnatal periods. Furthermore, whereas *Bmp7^R-GFlag/R-GFlag^* and *Bmp7^R-GFlag/+^;Bmp4^−/+^* mutants show stereotypical heart defects such as thinner myocardial walls, a common atrium, and a small, malformed OFT relative to wild type littermates (*Kim et al., 2019*), *Bmp4^S91C/S91C^* mutant hearts show grossly normal patterning of atria, ventricles and outflow tract while *Bmp4^−/E93G^* mutant hearts show reduced myocardial trabeculation and VSDs.

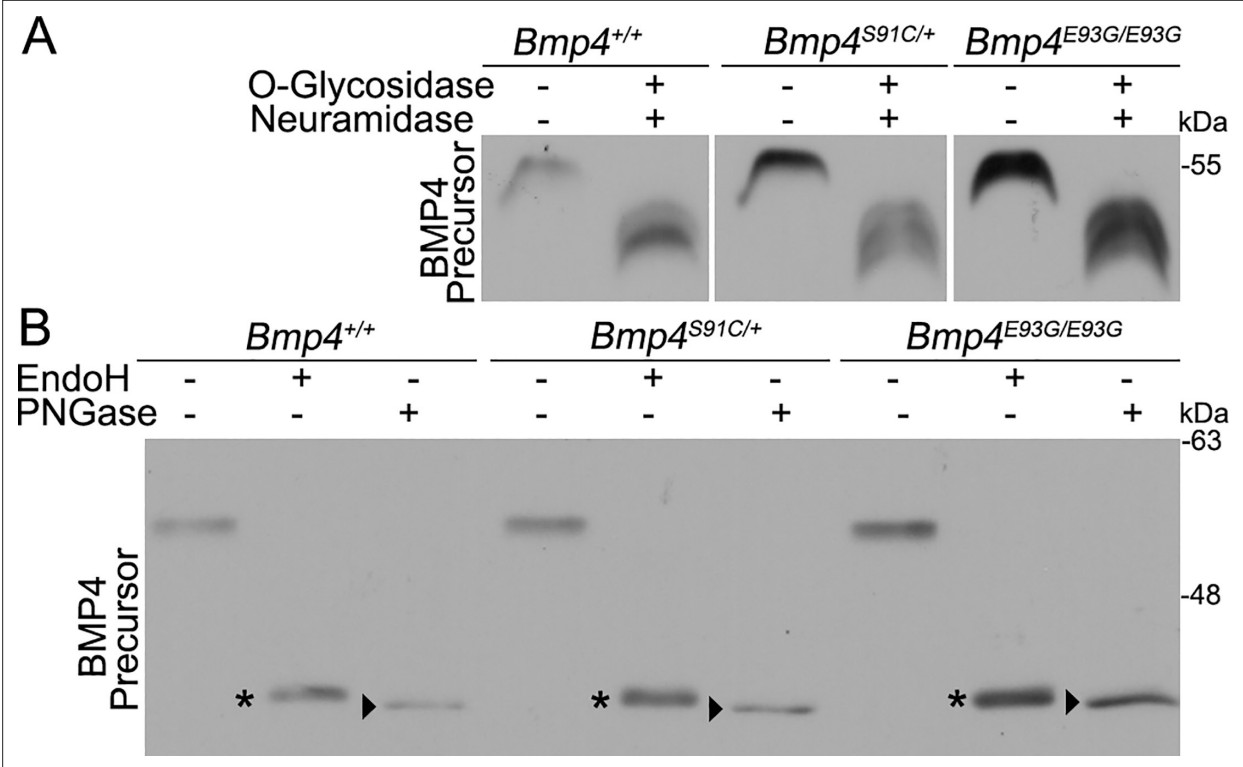

**Figure 7.** BMP4[E93G] and BMP4[S91C] precursor proteins are *O-* and *N-*glycosylated and exit the ER. (**A, B**) Mouse embryonic fibroblasts (MEFs) were isolated from E13.5 *Bmp4*[S91C] heterozygotes, *Bmp4*[E93G] homozygotes, or wild type littermates. Protein lysates were left untreated or were treated with *O-*glycosidase and α-neuramidase to remove *O-*linked glycosylation (**A**) or with EndoH or PNGase to remove *N-*linked glycosylation (**B**). Immunoblots were probed with antibodies directed against BMP4 to detect BMP4 precursor protein. Bands corresponding to Endo H-sensitive (asterisks) and Endo H-resistant PNGase-sensitive BMP4 (arrowheads) are indicated in B.

The online version of this article includes the following source data for figure 7:

**Source data 1.** TIFF files containing original western blots for *Figure 7A*, indicating the relevant bands and genotypes.

**Source data 2.** TIFF files containing original western blots for *Figure 7A*.

**Source data 3.** TIFF files containing original western blots for *Figure 7B*, indicating the relevant bands and genotypes.

**Source data 4.** TIFF files containing original western blots for *Figure 7B*.

The late lethality and heart defects observed in *Bmp4*[−/E93G] mutant mice phenocopy those observed in *Ngly1* mutant mice, which have a selective loss of BMP4 homodimers, but not heterodimers (*Fujihira et al., 2017*; *Galeone et al., 2020*). *Ngly1* and *Bmp4*[−/E93G] mutant mice both show VSDs, poorly elaborated myocardial trabeculation and late stage (P0) lethality (*Fujihira et al., 2017*; *Galeone et al., 2020*). NGLY1 is a deglycosylase that is required to clear misfolded BMP4 precursor monomers from the ER. This, in turn, promotes the formation of properly folded BMP4 homodimers that can be transported out of the ER to be cleaved to generate the functional ligand (*Galeone et al., 2020*). The *Drosophila* ortholog of NGLY1 (Pngl1) is required for formation of functional Dpp (the fly ortholog of BMP2/4) homodimers, but not heterodimers (*Galeone et al., 2017*). In *pngl1* mutant flies, and in MEFs from *Ngly1* mutant mice, Dpp/BMP4 precursor accumulates in the ER and very little cleaved ligand is observed (*Galeone et al., 2020*). By contrast, in MEFs from *Bmp4*[E93G/E93G] mutant mice BMP4 precursor accumulates in post-ER compartments although a similar loss of cleaved ligand is observed. The almost complete loss of mature BMP4 secreted from *Bmp4*[E93G/E93G] mutant MEFs is consistent with a selective loss of homodimers since *Bmp7* is not expressed in MEFs (*Lienert et al., 2011*) and thus BMP4 is predicted to exist primarily as a homodimer in these cells. By contrast, in E10.5 protein lysates BM4[E93G] appears to be cleaved normally, which may reflect a predominance of BMP4/7 heterodimers in early embryonic stages (*Kim et al., 2019*) or spatiotemporal differences in phosphorylation-dependent cleavage of BMP4 homodimers. In MEFs, mutations that prevent phosphorylation of the BMP4 prodomain, or mutations in *Ngly1* both lead to reduced levels of mature BMP4 homodimers

and similar phenotypic consequences, although the mechanisms underlying the loss of proteolytic maturation are distinct.

## Ideas and speculation

How might phosphorylation facilitate proteolytic maturation of BMP4 homodimers? We have previously shown that cleavage of BMP4 occurs in membrane proximal vesicles, and this is proposed to protect the ligand from degradation by passing it off to extracellular matrix binding partners as cleavage occurs (*Tilak et al., 2014*). This raises the question of how BMP4 can traffic through the TGN, where furin is first active (*Anderson et al., 2002*; *Thomas, 2002*), yet escape cleavage until it reaches the cell surface. We propose two possible mechanisms by which FAM20C-mediated phosphorylation facilitates cleavage of BMP4 and also ensures that this occurs in a membrane proximal subcellular compartment. One model proposes that phosphorylation occurs in the TGN and directs trafficking of BMP4 out of the Golgi to the cell surface via a route that is distinct from the pathway taken by furin (*Thomas, 2002*; *Figure 8A*). In this model, furin would first encounter and cleave BMP4 in membrane-proximal intracellular compartments, similar to what has been observed for other substrates (*Thomas, 2002*). This model predicts that BMP4$^{E93G}$ and BMP4$^{S91C}$ precursor proteins are trapped in the TGN or other subcellular compartments that furin cannot access. An alternate model proposes that furin, BMP4 and FAM20C traffic together out of the TGN but that the kinase is not active until it exits the TGN. In this model, phosphorylation in a membrane proximal compartment is required for furin to cleave BMP4 (*Figure 8B*). In support of this model, FAM20C is synthesized as a basally active kinase that is tethered to the Golgi by its propeptide. Propeptide cleavage by Site-I protease (S1P) releases Fam20C from the Golgi and greatly enhances its kinase activity as it to traffics to the cell surface (*Chen et al., 2021*). Furthermore, AlphaFold structure predictions position the phosphorylated serine of BMP4 (Ser91) in close proximity to the -R$^{287}$-R-R-A-K-R$^{292}$ sequence motif recognized by furin (*Figure 8C–E*; *Jumper et al., 2021*; *Varadi and Velankar, 2023*). This simple model building indicates the possibility of direct contact between pSer91 and Arg289, and that phosphorylation is required for furin to access the cleavage site, although we note that predictions surrounding the furin motif represent low probability conformations (*Figure 8—figure supplement 1*). Our observation that a putative phosphomimetic form of BMP4 (BMP4$^{S91D}$) generates reduced activity relative to wild type BMP4 when ectopically expressed in *Xenopus* embryosmight be interpreted to support the opposite conclusion, that phosphorylation inhibits, and dephosphorylation is required to promote access to the furin cleavage site. However, this conclusion is not compatible with the finding that two mutations associated with birth defects in humans (p.S91C or p.E93G), both of which are predicted to prevent FAM20C-mediated phosphorylation of the BMP4 prodomain, lead to impaired proteolytic maturation of endogenous BMP4 and reduced BMP activity in vivo. This model predicts that BMP4$^{E93G}$ and BMP4$^{S91C}$ precursor proteins can traffic to membrane proximal locations but remain uncleaved.

Protein structure predictions provide a possible explanation for why phosphorylation of the BMP4 prodomain may be required for cleavage of BMP4 homodimers, but not BMP4/7 heterodimers. Alphafold predicts that the furin cleavage motif on each monomeric chain of BMP4 homodimers (*Figure 8F*, purple) is situated on opposite sides of the dimeric molecule, each in close proximity to S91 (*Figure 8C, D*). By contrast, the two furin cleavage motifs (*Figure 8G*, purple) on heterodimers composed BMP4 (light shading) and BMP7 (dark shading) are predicted to be located on the same face of the dimeric molecule, in close proximity to each other (*Jumper et al., 2021*; *Varadi and Velankar, 2023*). This raises the possibility that furin can be recruited to the cleavage motif on BMP7, enabling it to access the cleavage motif on BMP4 independent of phosphorylation when present as a BMP4/7 heterodimer.

In summary, our data suggest that phosphorylation of the BMP4 prodomain is required for proteolytic maturation of BMP4 homodimers but not heterodimers, although further studies will be required to validate this in vivo and to understand the process mechanistically. Previous studies have demonstrated the importance of BMP heterodimers as endogenous ligands during early development of multiple organisms (*Bauer et al., 2023*; *Kim et al., 2019*; *Little and Mullins, 2009*; *Shimmi et al., 2005*). The current findings show that BMP4 homodimers are also functionally relevant but most likely operate primarily during later stages of organogenesis. The existence and relative role of heterodimers versus homodimers is likely to vary widely among different organisms, tissues, and developmental

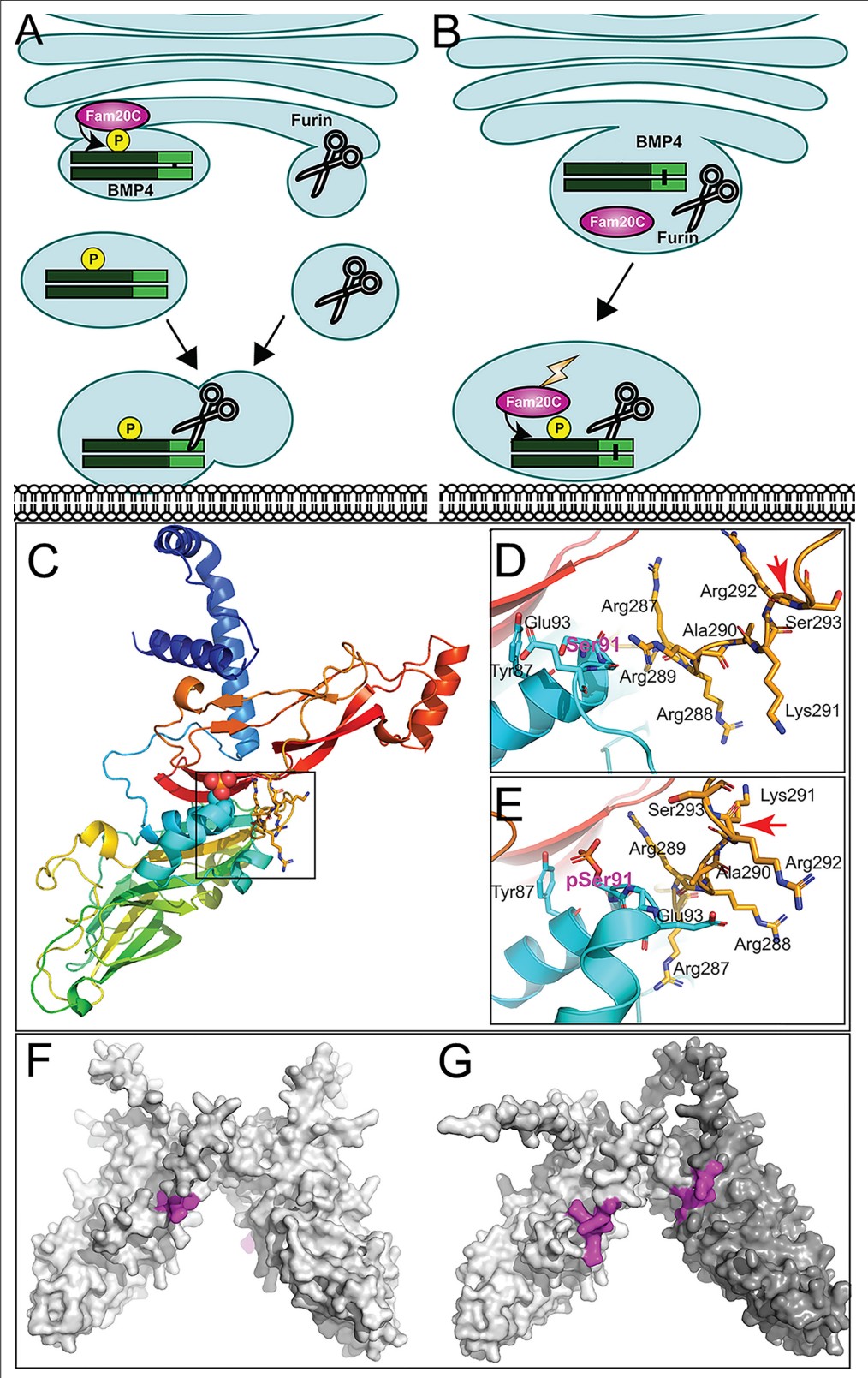

**Figure 8.** Hypothetical models for how prodomain phosphorylation regulates proteolytic maturation of BMP4. (**A**) Model 1: Phosphorylation of BMP4 by Fam20C within the trans-Golgi network (TGN) directs subcellular trafficking of BMP4 out of the Golgi to the cell surface via a route that is distinct from the pathway taken by furin. Furin- and BMP4-containing vesicles fuse in a region adjacent to the cell surface where furin cleaves BMP4

*Figure 8 continued on next page*

*Figure 8 continued*

to release the active ligand. (**B**) Model 2: Fam20C, BMP4, and furin traffic together to a membrane proximal subcellular compartment where Fam20C becomes catalytically activated (lightning bolt) and phosphorylates the prodomain of BMP4. Phosphorylation enables furin to access and cleave the consensus motif on BMP4. (**C**) Structure of BMP4 precursor monomer predicted by alphafold. N-terminus in blue, C-terminus in red. Boxed region includes Ser91 and the furin consensus motif. Close up of boxed region in C illustrating the close proximity between Ser91 (pink letters) within the prodomain in its unphosphorylated (**D**) or phosphorylated (**E**) form and the -R$^{287}$-R-R-A-K-R$^{292}$ furin consensus motif at the C-terminus of the prodomain. Red arrows indicate the site of furin cleavage. Spacefill representation of the of the structures of BMP4 precursor protein homodimers (**F**) or BMP4 (light gray) and BMP7 (dark gray) precursor protein heterodimers (**G**) predicted by alphafold with the location of furin recognition motifs denoted in purple.

The online version of this article includes the following figure supplement(s) for figure 8:

**Figure supplement 1.** Relative confidence in structural predictions across different regions of BMP4 and BMP7 precursor proteins.

stages. Additional studies are required to understand where and how a given BMP is directed to form one species of the other in vivo.

# Materials and methods

### Key resources table

| Reagent type (species) or resource | Designation | Source or reference | Identifiers | Additional information |
|---|---|---|---|---|
| Genetic reagent (*M. musculus*) | Bmp4−/+ | PMID:10049358 | RRID:MGI:2158495 | Dr. Brigid Hogan (Duke University) |
| Genetic reagent (*M. musculus*) | BRE-LacZ | PMID:15331632 | | Dr. Christine Mummery (Leiden University) |
| Genetic reagent (*M. musculus*) | Bmp4$^{S91C}$ | This paper | | Generated using CRISPR–Cas9 technology |
| Genetic reagent (*M. musculus*) | Bmp4$^{E93G}$ | This paper | | Generated using CRISPR–Cas9 technology |
| Recombinant DNA reagent | CS2+mouse BMP4HAMyc | PMID:15356272 | | Catherine Degnin (Oregon Health and Science University) |
| Recombinant DNA reagent | CS2+mouse BMP7Flag | PMCID:PMC6785266 | | Hyungseok Kim (University of Utah) |
| Recombinant DNA reagent | CS2+mouse BMP4HAMycS91C | This paper | | Site-directed mutagenesis (Agilent) |
| Recombinant DNA reagent | CS2+mouse BMP4HAMycS91D | This paper | | Site-directed mutagenesis (Agilent) |
| Recombinant DNA reagent | CS2+mouse BMP4HAMycE93G | This paper | | Site-directed mutagenesis (Agilent) |
| Antibody | Rabbit polyclonal anti-pSmad1/5 | Cell Signaling, Cat. #9511S | RRID:AB_491015 | WB (1:1000) |
| Antibody | Mouse monoclonal anti-BMP4 | Santa Cruz | RRID:AB_2063534 | WB (1:1000) |
| Antibody | Rabbit polyclonal anti-beta actin | Sigma | RRID:AB_476693 | WB (1:10,000) |
| Antibody | HRP-conjugated anti-rabbit polyclonal IgG | Jackson Immuno Research | Cat. # 111-035-144 | WB (1:10,000) |
| Antibody | HRP-conjugated anti-mouse polyclonal IgG2b | Jackson Immuno Research | Cat. # 115-035-207 | WB (1:10,000) |
| Commercial assay or kit | Pierce ECL western blotting substrate | Fisher | Cat. # 32209 | |
| Commercial assay or kit | BCA Protein Assay Kit | Fisher | Cat. # 23225 | |
| Chemical compound, drug | Halt protein and phosphatase inhibitor | Fisher | Cat. # 78442 | |
| Commercial assay or kit | QuickChange II XL site-directed mutagenesis kit | Agilent | Cat. # 20052 | |

### *Xenopus* embryo culture and manipulation

This study was performed in strict accordance with the recommendations in the Guide for the Care and Use of Laboratory Animals of the National Institutes of Health. All of the animals were handled according to approved Institutional Animal Care and Use Committee (IACUC) protocol (#00001610) of the University of Utah. Every effort was used to minimize stress and suffering. Embryos were obtained,

microinjected, and cultured as described (*Mimoto and Christian, 2011*). Embryo explants were performed as described (*Mimoto et al., 2015*; *Neugebauer et al., 2015*).

## Mouse strains

This study was performed in strict accordance with the recommendations in the Guide for the Care and Use of Laboratory Animals of the National Institutes of Health. Animal procedures followed protocols approved by the University of Utah Institutional Animal Care and Use Committees (#00001548). Every effort was used to minimize stress and suffering. $Bmp4^{LacZ/+}$ (RRID:MGI:3811252) and BRE-LacZ mice were obtained from Dr. B. Hogan (Duke University) and Dr. C. Mummery (Leiden University), respectively. $Bmp4^{S91C}$ and $Bmp4^{E93G}$ mice were generated by personnel in the Mutation Generation & Detection and the Transgenic & Gene Targeting Mouse Core Facilities at the University of Utah using CRISPR–Cas9 technology. sgRNA RNAs (5′-TGAGCTCCTGCGGGACTTCG-3′) were injected into C57BL/6J zygotes together with long single stranded donor DNA repair templates (E93G: 5′-GGAAGAAAAAAGTCGCCGAGATTCAGGGCCACGCGGGAGGACGCCGCTCAGGGCAGAGCC ATGAGCTCCTGCGGGACTTCGAaGCGACAC<u>TTTATCCATACGACGTGCCAGACTATGCA</u>CTAC AGATGTTTGGGCTGCGCCGCCGTCCGCAGCCTAGCAAGAGCGCCGTCATTCCGGATTACATGAG GGATCTTTACCGGCTCCAGTCTGG**a**G**g**GGAGGAGGAGGAAGAGCAGAGCCAGGGAACCGGGCT TGAGTACCCGGAGCGTCCCGCCAGCCGAGCCAACACTG-3′; S91C: 5′- GGAAGAAAAAAGTCGC CGAGATTCAGGGCCACGCGGGAGGACGCCGCTCAGGGCAGAGCCATGAGCTCCTGCGGGA CTTCGAaGCGACAC<u>TTTATCCATACGACGTGCCAGACTATGCA</u>CTACAGATGTTTGGGCTGCG CCGCCGTCCGCAGCCTAGCAAGAGCGCCGTCATTCCGGATTACATGAGGGATCTTTACCGGCTC CAGT**g**TGG**c**GAGGAGGAGGAGGAAGAGCAGAGCCAGGGAACCGGGCTTGAGTACCCGGAGC GTCCCGCCAGCCGAGCCAACACTG-3′; sequence encoding HA epitope underlined; nucleotide changes bold and small case) and Cas9 protein. G0 founders were crossed to C57BL/6J females to obtain heterozygotes. DNA fragments PCR-amplified from genomic DNA were sequenced to verify the presence of the epitope tag and absence of other sequence changes. Genotypes were determined by PCR amplification of tail DNA using primers that anneal to sequence immediately surrounding the HA epitope tag (5′ primer: 5′-TATGCCAAGTCCTGCTAG-3′ and 3′ primer: 5′-GATC CCTCATGTAATCCG-3′) under the following conditions: 94°C for 30 s, 60°C for 30 s, 72°C for 30 s, 35 cycles. cDNA constructs cDNAs encoding mouse BMP4$^{HAMyc}$ and BMP7$^{Flag}$ have been described previously (*Kim et al., 2019*; *Tilak et al., 2014*). cDNAs encoding BMP4$^{HAMycS91C}$, BMP4$^{HAMycS91D}$, and BMP4$^{HAMycE93G}$ were generated using a QuickChange II XL site-directed mutagenesis kit (Agilent Technologies).

## Analysis of RNA

Total RNA was isolated using TRIzol (Invitrogen). Semi quantitative RT-PCR was performed as described (*Nakayama et al., 1998*) using primers specific for *odc* or tbxt (*Supplementary file 3*) and an annealing temperature of 58°C.

## In situ hybridization and β-galactosidase staining

Embryos were processed for in situ hybridization with digoxigenin-labeled Nkx2.5 riboprobes as described previously (*Wilkinson and Nieto, 1993*). β-Galactosidase staining of BRE-LacZ embryos was performed as described (*Lawson et al., 1999*) and color was developed using RedGal. Investigators were blinded to genotype until after morphology and/or staining intensity had been documented.

## Immunoblot analysis of *Xenopus* extracts

Proteins were harvested from 10 pooled DMZ explants by freon extraction as described previously (*Mimoto and Christian, 2011*). Proteins were resolved by SDS–PAGE under reducing and non-reducing conditions and transferred onto PVDF membranes. Membranes were probed with anti-actin (1:10,000, Sigma, *RRID*:AB_476693) and anti-pSmad1/5 (1:1000, Cell Signaling *RRID*:AB_491015) antibodies. Immunoreactive proteins were detected using Enhanced Chemiluminescence reagent (Pierce) and light emissions captured with x-ray film. Images were scanned and relative band intensity was quantified using ImageJ software.

## Immunoblot analysis of mouse embryo lysates and MEFs

Mouse embryos were dissected from pregnant females at E10.5, homogenized in lysis buffer 150 mM NaCl, 20 mM Tris-Cl pH 7.5, 1 mM EDTA, 1% sodium deoxycholate, 1% NP40, 1× protease inhibitor (cOmplete Mini, EDTA-free, Roche), 1× phosphatase inhibitor cocktail (Sigma), and protein concentration was measured by BCA kit (Thermo Scientific). 80, 10, and 5 µg of embryo lysates were used for detection of BMP4 mature ligand, BMP4 precursor, and pSmad1, respectively. MEFs were isolated at E13.5 as described (*Durkin et al., 2013*) and cultured in 10% FBS, 1× pen/strep, 1× Glutamate in DMEM. Following the second passage, MEFs were cultured in serum containing media until they reached 80% confluency and were then cultured in serum-free media for 24 hr before collecting cells and conditioned media. 800 µl of conditioned media was used for TCA precipitation. Proteins were deglycosylated according to the manufacturer's instructions using *O*-glycosidase, neuramidase, EndoH and PNGase purchased from NEB. Proteins were separated by electrophoresis on 10% or 12% gels and transferred to PVDF membranes that were probed with anti-Bmp4 (1:1000, Santa Cruz RRID:AB_2063534), anti-pSmad1/5 (1:1000, Cell Signaling, RRID:AB_491015), or anti-actin (1:10,000, Sigma, RRID:AB_476693) antibodies followed by HRP-conjugated anti-rabbit IgG or anti-mouse IgG2b (Jackson ImmunoResearch) secondary antibodies. Immunoreactive proteins were detected using Enhanced Chemiluminescence reagent (Pierce) and light emissions captured with x-ray film. Images were scanned and relative band intensity was quantified using ImageJ software.

## Structure prediction

Full-length (starting after processed signal sequence) Bmp4 homodimer and Bmp4/7 heterodimer sequences were input into https://alphafoldserver.com/ running AlphaFold version 3. Protein structures were analyzed and figures made using PyMol (*Schrodinger LLC, 2020*) (retrieved from http://www.pymol.org/pymol). Quality of prediction as measured by the pLDDT can be observed per residue in Figure S7 where very low (<50), low (50–70), confident (70–90), and very high (>90) rankings are shown indicating that the furin binding loops are of low predictive confidence.

## Statistics

NIH ImageJ software was used to quantify band intensities. A Student's *t*-test was used to compare differences in gene expression or protein levels between two groups. Differences with p < 0.05 were considered statistically significant. All results were reproduced in three biological replicates.

## Materials availability statement

All materials used in this study are available upon request. Researchers may contact Dr. Jan Christian (jan.christian@neuro.utah.edu) for access. Most materials have no restrictions; however, access to the mouse model may require a Material Transfer Agreement (MTA) to protect intellectual property or for regulatory compliance. Dr. Christian can provide details on terms and facilitate the transfer process.

# Acknowledgements

We thank Dr. Eyad Marashly for generating the BMP4$^{E93G}$ cDNA and synthetic RNAs and Dr. Diana Lim for education in the use of Adobe illustrator. We would like to acknowledge the University of Utah Mutation Generation and Detection Core and the Transgenic and Gene Targeting Core for generating the *Bmp4*$^{S91C}$ and *Bmp4*$^{E93G}$ mouse lines. This work utilized DNA and peptide shared resources supported by the Huntsman Cancer Foundation and the National Cancer Institute of the NIH (grant P30CA042014). The content is solely the responsibility of the authors and does not represent the official views of the NIH. This work was supported by the Eunice Kennedy Shriver National Institute of Child Health and Human Development, Grant/Award Numbers: R21HD102668; R21HD102668-W1 to JLC; National Institute of Diabetes and Digestive and Kidney Diseases, Grant/Award Number: R01DK128068 to JLC.

# Additional information

### Funding

| Funder | Grant reference number | Author |
|---|---|---|
| Eunice Kennedy Shriver National Institute of Child Health and Human Development | R21HD102668 | Jan L Christian |
| Eunice Kennedy Shriver National Institute of Child Health and Human Development | R21HD102668-W1 | Jan L Christian |
| National Institute of Diabetes and Digestive and Kidney Diseases | R01DK128068 | Jan L Christian |

The funders had no role in study design, data collection, and interpretation, or the decision to submit the work for publication.

### Author contributions

Hyung-Seok Kim, Conceptualization, Data curation, Formal analysis, Supervision, Validation, Investigation, Visualization, Methodology, Project administration, Writing – review and editing; Mary L Sanchez, Resources, Formal analysis, Validation, Investigation, Visualization, Methodology, Writing – review and editing; Joshua Silva, Resources, Data curation, Formal analysis, Validation, Investigation, Visualization, Methodology, Writing – review and editing; Heidi L Schubert, Data curation, Software, Formal analysis, Validation, Methodology, Writing – review and editing; Rebecca Dennis, Investigation, Visualization, Methodology; Christopher P Hill, Software, Supervision, Validation, Writing – review and editing; Jan L Christian, Conceptualization, Resources, Data curation, Formal analysis, Supervision, Funding acquisition, Validation, Investigation, Visualization, Methodology, Writing - original draft, Project administration, Writing – review and editing

### Author ORCIDs

Mary L Sanchez ⓘ https://orcid.org/0009-0005-5386-283X
Joshua Silva ⓘ https://orcid.org/0009-0005-6162-9966
Heidi L Schubert ⓘ https://orcid.org/0000-0003-1762-8390
Christopher P Hill ⓘ https://orcid.org/0000-0001-6796-7740
Jan L Christian ⓘ https://orcid.org/0000-0003-3812-3658

### Ethics

This study was performed in strict accordance with the recommendations in the Guide for the Care and Use of Laboratory Animals of the National Institutes of Health. All of the animals were handled according to approved Institutional Animal Care and Use Committee (IACUC) protocols (#00001610; Xenopus and #00001548; mouse) of the University of Utah. Every effort was used to minimize stress and suffering.

Reviewer #1 (Public review): https://doi.org/10.7554/eLife.105018.4.sa1
Reviewer #2 (Public review): https://doi.org/10.7554/eLife.105018.4.sa2
Reviewer #3 (Public review): https://doi.org/10.7554/eLife.105018.4.sa3
Author response https://doi.org/10.7554/eLife.105018.4.sa4

# Additional files

### Supplementary files

Supplementary file 1. Progeny from *Bmp4*$^{S91C/+}$ intercrosses at embryonic stages by sex.
Supplementary file 2. Progeny from Bmp4E93G/+ x Bmp4-/+ crosses at embryonic stages by sex.
Supplementary file 3. Primers used for PCR.

MDAR checklist

## Data availability

All data generated or analyzed during this study are included in the manuscript and supporting files.

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
