## [Editor Report · eLife Assessment]

This **fundamental** work presents two clinically relevant BMP4 mutations that contribute to vertebrate development. The **compelling** evidence, both from wet lab and AI generated predictions, supports that the site-specific cleavage at the BMP4 pro-domain precisely regulates its function and provides mechanistic insight how homodimers and heterodimers behave differently. The work will be of broad interest to researchers working on growth factor signaling mechanisms and vertebrate development.

---

## [Referee Report · Reviewer #1 (Public review)]

Summary:

The authors demonstrate that two human preproprotein human mutations in the BMP4 gene cause a defect in proprotein cleavage and BMP4 mature ligand formation, leading to hypomorphic phenotypes in mouse knock-in alleles and in *Xenopus embryo* assays.

Strengths:

They provide compelling biochemical and in vivo analyses supporting their conclusions, showing the reduced processing of the proprotein and concomitant reduced mature BMP4 ligand protein from impressively mouse embryonic lysates. They perform excellent analysis of the embryo and post-natal phenotypes demonstrating the hypomorphic nature of these alleles. Interesting phenotypic differences between the S91C and E93G mutants are shown with excellent hypotheses for the differences. Their results support that BMP4 heterodimers act predominantly throughout embryogenesis whereas BMP4 homodimers play essential roles at later developmental stages.

Weaknesses:

In the revision the authors have appropriately addressed the previous minor weaknesses.

---

## [Referee Report · Reviewer #2 (Public review)]

Summary:

The revised paper by Kim et al. reports two disease mutations in proBMP4, S91C and E93G, disrupt the FAM20C phosphorylation site at Ser91, blocking the activation of proBMP4 homodimers, while still allowing BMP4/7 heterodimers to function. Analysis of DMZ explants from *Xenopus embryos* expressing the proBMP4 S91C or E93G mutants showed reduced expression of pSmad1 and tbxt1. The expert amphibian tissue transplant studies were expanded to in vivo studies in Bmp4S91C/+ and Bmp4E93G/+ mice, highlighting the impact of these mutations on embryonic development, particularly in female mice, consistent with patient studies. Additionally, studies in mouse embryonic fibroblasts (MEFs) demonstrated that the mutations did not affect proBMP4 glycosylation or ER-to-Golgi transport but appeared to inhibit the furin-dependent cleavage of proBMP4 to BMP4. Based on these findings and AI modeling using AlphaFold of proBMP4, the authors speculate that pSer91 influences access of furin to its cleavage site at Arg289AlaLysArg292 in a new "Ideas and Speculation" section. Overall, the authors addressed the reviewers' comments, improving the presentation.

Strengths:

The strengths of this work continue to lie in the elegant *Xenopus* and mouse studies that elucidate the impact of the S91C and E93G disease mutations on BMP signaling and embryonic development. Including an "Ideas and Speculation" subsection for mechanistic ideas reduces some shortcomings regarding the analysis of the underlying mechanisms.

---

## [Referee Report · Reviewer #3 (Public review)]

Summary:

The authors describe important new biochemical elements in the synthesis of a class of critical developmental signaling molecules, BMP4. They also present a highly detailed description of developmental anomalies in mice bearing known human mutations at these specific elements.

Strengths:

This paper presents exceptionally detailed descriptions of pathologies occurring in BMP4 mutant mice. Novel findings are shown regarding the interaction of propeptide phosphorylation and convertase cleavage, both of which will move the field forward. Lastly, a provocative hypothesis regarding furin access to cleavage sites is presented, supported by Alphafold predictions.

---

## [Author Response]

The following is the authors’ response to the previous reviews

**Reviewer #2 (Public review):**
Summary:The revised paper by Kim et al. reports two disease mutations in proBMP4, S91C and E93G, disrupt the FAM20C phosphorylation site at Ser91, blocking the activation of proBMP4 homodimers, while still allowing BMP4/7 heterodimers to function. Analysis of DMZ explants from *Xenopus embryos* expressing the proBMP4 S91C or E93G mutants showed reduced expression of pSmad1 and tbxt1. The expert amphibian tissue transplant studies were expanded to in vivo studies in Bmp4S91C/+ and Bmp4E93G/+ mice, highlighting the impact of these mutations on embryonic development, particularly in female mice, consistent with patient studies. Additionally, studies in mouse embryonic fibroblasts (MEFs) demonstrated that the mutations did not affect proBMP4 glycosylation or ER-to-Golgi transport but appeared to inhibit the furin-dependent cleavage of proBMP4 to BMP4. Based on these findings and AI modeling using AlphaFold of proBMP4, the authors speculate that pSer91 influences access of furin to its cleavage site at Arg289AlaLysArg292 in a new "Ideas and Speculation" section. Overall, the authors addressed the reviewers' comments, improving the presentation.Strengths:The strengths of this work continue to lie in the elegant *Xenopus* and mouse studies that elucidate the impact of the S91C and E93G disease mutations on BMP signaling and embryonic development. Including an "Ideas and Speculation" subsection for mechanistic ideas reduces some shortcomings regarding the analysis of the underlying mechanisms.Weaknesses:(1) (Minor) In Figure S1 and lines 165-174 and 179-180, the authors should consider that, unlike the wild-type protein (Ser), which can be reversibly phosphorylated or dephosphorylated, phosphomimic mutations are locked into mimicking either the phosphorylated state (Asp) or the non-phosphorylated state (Ala). Consequently, if the S91D mutant exhibits lower activity than WT, it could imply that S91D interferes with other regulatory constraints, as the authors suggest. However, it may also be inhibiting activation. Therefore, caution is warranted when comparing S91D with S91C to conclude that Ser91 phosphorylation increases BMP4 activity. While additional experiments are not necessary, further consideration is essential.(Minor) In lines 394-399, the authors cleverly speculate that pS91 interacts with Arg289-the essential P4 arginine for furin processing. If so, this interaction could hinder the cleavage of proBMP4, as indicated by the results in Figure S1. The discussion would benefit from considering that, contrary to their favored model, dephosphorylation at Ser91 might actually facilitate cleavage.

We have added a paragraph raising this possibility but explaining why it is unlikely and inconsistent with our in vivo data. The S91D construct was a simple control that was tested in ectopic expression assays and not in vivo. We can make no conclusions about whether this construct resembles the phosphorylated state or whether it hinders or facilitates cleavage in vivo. The conclusion that dephosphorylation promotes BMP4 cleavage or activity is not compatible with the finding that two mutations associated with birth defects in humans (p.S91C or p.E93G) that are predicted to prevent FAM20C-mediated phosphorylation of the BMP4 prodomain lead to impaired proteolytic maturation of endogenous BMP4 and reduced BMP activity in vivo.

(2) In Figure 4, panels A, E, and I, the proBMP bands in the mouse embryonic lysates and MEFs expressing the mutations show a clear size shift. Are these shifts a cause or a consequence of the lack of cleavage? Regardless, the size shifts should be explicitly noted.

These intriguing shifts were observed in some but not all biological replicates. When present, the shifts were not reversed by treatment with phosphatases or deglycosylases, and the shifts were never observed in epitope tagged wild type controls. We have added a paragraph noting the shifts and our tests of whether they might be due to glycosylation, phosphorylation or epitope tags.

(3) (Minor) In line 314, the authors should consider modifying the wording to: "is required for modulating proprotein convertase..."

The original wording (“Collectively, our findings are consistent with a model in which FAM20C-mediated phosphorylation of the BMP4 prodomain is not required for folding or exit of the precursor protein from the ER, but is required for proprotein convertase recognition and/or for trafficking to post-TGN compartment(s) where BMP4 is cleaved”) more accurately reflects the model that is supported by our findings. Stating that “phosphorylation ……is required to modulate proprotein convertase recognition and/or trafficking” is vague and leaves open the possibility that it modulates in either direction, which our data do not support as described in point 1 above.